# Neutrophils and Anesthetic Drugs: Implications in Onco-Anesthesia

**DOI:** 10.3390/ijms25074033

**Published:** 2024-04-04

**Authors:** Alexandru Leonard Alexa, Sergiu Sargarovschi, Daniela Ionescu

**Affiliations:** 1Department of Anesthesia and Intensive Care I, “Iuliu Haţieganu” University of Medicine and Pharmacy, 400012 Cluj-Napoca, Romania; sergiu.sargarovschi@gmail.com (S.S.); daniela_ionescu@umfcluj.ro (D.I.); 2Association for Research in Anesthesia and Intensive Care (ACATI), 400162 Cluj-Napoca, Romania; 3Onco-Anaesthesia Research Group, ESAIC, 1000 Brussels, Belgium; 4Outcome Research Consortium, Cleveland, OH 44195, USA

**Keywords:** neutrophil, anesthesia, neutrophil extracellular traps, neutrophil-to-lymphocyte ratio, propofol, opioids, local anesthetics

## Abstract

Apart from being a significant line of defense in the host defense system, neutrophils have many immunological functions. Although there are not many publications that accurately present the functions of neutrophils in relation to oncological pathology, their activity and implications have been studied a lot recently. This review aims to extensively describe neutrophils functions’; their clinical implications, especially in tumor pathology; the value of clinical markers related to neutrophils; and the implications of neutrophils in onco-anesthesia. This review also aims to describe current evidence on the influence of anesthetic drugs on neutrophils’ functions and their potential influence on perioperative outcomes.

## 1. Introduction

Neutrophils (also known as polymorphonuclear leukocytes—PMNs) represent 50–70% of circulating leukocytes [1,2]. They are produced in bone marrow from where they enter the circulation and, subsequently, migrate to the tissues where they complete their functions [3]. PMNs can rapidly move to the tissues at the site of infection and inflammation via the leukocyte adhesion cascade, a complex cascade of adhesive steps between leukocytes and the endothelium [4,5].

Neutrophils are the first line of defense of the innate immune system against microorganisms [2,6], and this is accomplished by carrying out multiple antimicrobial functions [3,7,8].

The general functions of neutrophils include phagocytosis, degranulation, the production of reactive oxygen species (ROS) and the formation of neutrophil extracellular traps (NETs) [2,3,9,10,11,12]. More recently, numerous studies have reported additional specialized functions of neutrophils, namely producing cytokines and other inflammatory markers that contribute to the regulation of immune system functions [2], mediation/resolution of inflammation, and cytokine/chemokine modulation of immunity [13].

In the last few years, it has been shown that anesthetic agents may influence both the number and the subpopulations of neutrophils and their functions, so it is important to briefly describe these functions before describing the implications of anesthetic agents on neutrophils’ functions, leading to potential impacts on a patient’s outcome.

## 2. Neutrophils’ Functions

### 2.1. Phagocytosis

Phagocytosis is the process by which a particle is internalized by neutrophils into a phagosome where most of the pathogens are killed [2,14,15]. This process is receptor-mediated, during which pathogen-associated molecular patterns (PAMPs) are implicated in the recognition of pathogens [2,14] due to specific receptors for PAMPs on neutrophils’ surface [2,15].

### 2.2. Degranulation

Degranulation is the process during which effector molecules stored in internal granules of neutrophils destroy bacteria/pathogens [2]. These internal granules are classified as primary, secondary, and tertiary and contain myeloperoxidase (MPO), azuracidin, and defensins (primary granules, the first to be produced during the neutrophil development–promyelocyte phase), metalloproteinase (MMP)-9, lactoferrin (included in the secondary granules), albumin, and cytokines [16], and they are highly cytotoxic (as secretory granules in PMN) [2].

Circulating neutrophils are, normally, in a resting state when they cannot degranulate, leading to tissue damage. Neutrophils’ degranulation requires priming with the help of bacterial pathogens or products, pro-inflammatory cytokines, and chemokines, leading to neutrophils being ready to degranulate when in contact with different stimuli like pathogen-associated molecular patterns (PAMPs) and damage associate patterns (DAMPs) [2,13,17,18]. 

### 2.3. ROS

The production of reactive oxygen species (ROS) by nicotinamide adenine dinucleotide phosphate oxidase (NADPH-oxidase) in neutrophils plays an important role in the antimicrobial activity of these cells against bacterial and fungal infections and in inflammation [19]; they also prevent the excessive degranulation of tertiary granules of neutrophils [2,20]. ROS deficiency leads to recurrent severe bacterial infections [19,21,22], while excessive ROS formation leads to excessive inflammation and SIRS (e.g., during COVID-19 [23]). ROS have a direct antimicrobial activity, killing bacteria and fungi and some types of viruses [24], but they can also cross the bacterial cell membrane, damaging nucleic acids, cell membrane and cell proteins [19,25].

Once produced, superoxide anions (O_2_^−^) undergo dismutation to hydrogen peroxide (H_2_O_2_) that, subsequently, oxidizes ferrous iron and generates highly reactive hydroxyl radical OH^−^. On the other hand, MPO released from PMN granules converts hydrogen peroxide into a hypochlorous acid (HOCl) that is highly bactericidal [19,26]. ROS have the capacity to decrease bacterial growth during infections due to their ability to diffuse through bacterial membranes and may also damage their DNA, protein, and lipid molecules; ROS may be also involved in inflammation due to infection [26].

It is interesting to note that bacteria have developed intrinsic and extrinsic resistance mechanisms to ROS by developing scavenger mechanisms [19,26] and through iron sequestration and DNA damage repair [19,26].

It is also of interest that some anesthetic drugs, like lidocaine, may interfere with ROS production or their tissue effects [27]. Similarly, vitamin C and vitamin E act as ROS eliminating substances.

### 2.4. Neutrophil Extracellular Traps (NETs)

#### 2.4.1. Formation of NETs

NETs are web-like networks of chromatin decondensed fibers covered with histones, enzymes from the granules (e.g., myeloperoxidase (MPO) and neutrophil elastase (NE) that can be used, along with citrullinated histone 3 (CitH3), as indirect markers of NETosis) [28,29] and antimicrobial proteins formed after nuclear membrane rupture, followed by the release of nucleoplasm into the cell and, finally, cell membrane destruction [3,30]. This network has a role in bacterial and cancer cell trapping and, finally, killing these cells after their removal from circulation [30].

The process of NET formation—NETosis—is defined as ROS-dependent regulated cell death [31]. However, progress has been made in describing the NETosis process, so it is actually considered to be a process that may be NADPH-oxidase (NOX)-dependent (former suicidal NETosis) and NOX-independent (so-called vital NETosis) [30,32,33], subsequently divided into three types [30]. The NOX-dependent pathway is triggered by lipopolysaccharide (LPS), phorbol 12-myristate 13-acetate (PMA), and bacteria, resulting in nuclear breakdown via ROS generation and the activation of certain enzymes like ERK1, P38, and Src [30,33]. On the other hand, the NOX-independent pathway is activated by bacteria and cancer cells, leading to the formation of ROS that act on certain enzymes, leading, in the end, to chromatin decondensation [30,34].

#### 2.4.2. Assessment of NET Formation

Several methods are described for the visualization and quantification of NETs. They are classified as direct and indirect methods.

Immunofluorescence microscopy is a direct qualitative method consisting of granular staining for neutrophil-associated enzymes, histones, and extracellular DNA, using specific dyes and immunostaining for the enzymatic components of NETs [35,36]. This method allows NET identification in cell cultures, tissue sections, peripheral blood, or bronchoalveolar lavage. It can differentiate between the different stages of NET formation and between NET apoptosis and necrosis [35]. However, it is operator-dependent, time-consuming, laborious, and does not allow for quantitative analysis [36]. Nowadays, a semi-automatic quantitative assessment of NETosis is possible by using specific antibodies against histones, chromatin, and the cleavage site of the histone H3 [29,35].

Electron microscopy and scanning electron microscopy (SEM). Electron microscopy and SEM consist of an electron beam passing through or reflecting from the surface of the sample [35], which is then projected onto a fluorescent surface is collected by a sensor that creates an image [37]. SEM is first choice for visualizing NET formation. It creates a better image compared with simple electron microscopy [29]. The disadvantage is determined by the possibility of the electron beam stimulating the formation of NETs [29].

In vivo and in situ methods use intravital two-photon microscopy and subsequent electron emission detection based on DNA components [35,36]. Recent data show that NETs are less expressed in vivo compared to in vitro [29].

Real-time NETosis assessment uses the IncuCyte ZOOM imaging platform (Wilmington, DE, USA). This platform uses cell membrane properties with the permeability of DNA dyes from NUCLEAR-ID (Enzo Life Sciences, Inc., Farmingdale, NY, USA) and Sytox Green (Thermo Fisher cat no S7020, Waltham, MA, USA). By detecting morphological changes in cells and nuclei, this method can identify NETs from other cell deaths but has the disadvantage of generating false-positive results if the cell membrane integrity is compromised [38].

Flow cytometry and multispectral image flow cytometry allows the measurement of scattered light or fluorescence signals emitted by properly irradiated cells, leading to the differentiation of distinct cell populations in whole blood. It uses specific dyes (Sytox Green) or conjugated antibodies, which bind to cytokines, receptors, cellular components, and DNA [35]. Multispectral image flow cytometry can measure in whole blood both “suicidal” and “vital” NETs, and it can quantify NETs in the early stages that underestimate the formation of NETosis at a longer time interval [29,35].

Enzyme-linked immunosorbent assay (ELISA) quantifies some products of neutrophil degradation characteristic of NETs without direct visualization [29,35]. However, there is controversy regarding the best marker for detecting NET formation [35]. The CitH3-DNA ELISA is validated for reliable NET quantification in human plasma. While MPO-DNA complexes may also be identified via ELISA, discrepancies between in vivo and in vitro values have been identified [35].

Western blot is another indirect method, using specific antibodies to quantify a protein involved in the formation of NETs. It is a very sensitive and specific method with some disadvantages including costs and the looseness of specificity in a mix of protein antibodies [35].

#### 2.4.3. Clinical Implications of NETs

##### NETs in Critical Patients

Although, as shown before, NETs contribute to bacterial clearance, excessive NETs promote inflammation and tissue injury in sepsis and other inflammatory diseases [39]. Thus, high levels of NET formation have been described for the bacterial origin (pneumonia) of ARDS or ventilation pneumonia [40], while there is a correlation between the severity of ARDS and NET serum level [41], as well as between the NET (evaluated by MPO-DNS complex) level and the severity of organ dysfunction and 28-day mortality in sepsis and septic shock [39,40]. Higher levels of NETs have been described during COVID-19, where NETs act as triggers of a cascade of inflammatory reactions, leading to a cytokine storm that is a characteristic of this viral infection [42,43]. NETs are also increased in cardiovascular diseases especially in atherosclerosis, venous thrombosis, and inflammatory pathologies like pancreatitis [44].

##### NETs and Cancer and Cancer Associated Thrombosis

Tumor progression depends on the balance between the inflammatory response generated by tumor-infiltrating immune cells and tumor-derived factors in the tumor microenvironment (TME). Neutrophils are the first cells to reach the inflammation/injury site. Recent data show that inflammation is closely linked with cancer development and that NETs generated by neutrophils in TME play a critical role in tumor development, progression, metastasis, and resistance to therapy [45,46,47].

This is why, more recently, NETs (or their indirect markers) have been considered biomarkers for different types of cancers [46]; elevated serum levels of NETs may be associated with cancer diagnosis [30,32,33]. It was demonstrated that high levels of CitH3 are associated with a poor prognosis in cancer patients [46,48,49]. The pathophysiology of NET formation in TME has a starting point in tumor cell necrosis generated by the hypoxic TME repleted with cytokines. DAMPs are, thus, released, causing inflammation and triggering NET formation, along with the activation of ROS pathways [30,45,46,50]. Studies have also showed that NETs are key players in transformation from inflammation to carcinogenesis by awakening dormant tumor cells and that the accumulation of intertumoral NETs facilitates tumor growth [46,50].

NETs are also implicated in tumor metastasis and cancer-associated thrombosis. It has been shown that NET-bound cancer cells, thus, promote metastasis [3,44,51]. It has also been shown that in certain types of cancers (breast, lung), metastases are associated with high levels of NETs [44,52,53,54]. Other implicated mechanisms are NET-DNA receptors on the tumor cell membrane [54]; cathepsin C, a protease released by tumor cells that primes NET formation, and the release of G-CSF at sites of NET dissemination and interleukin disbalance [44,54,55].

NETs are also associated with venous thromboembolism—common in most cancers— or a pro-thrombotic state in different cancers, like, for example, pancreatic cancer, typically associated with a hypercoagulability state, responsible for venous thromboembolism [44,52].

## 3. Cytokine Release

In the last few years, it has been shown that neutrophils’ release stimulates pro- (IL-6, TNF-α, IL-1, IL-17, IFN-α and γ) and anti-inflammatory interleukins (IL-4, IL-10, etc.), as well as other chemokines (CXCL1, CXCL8, CXCL10, CCL2-4), angiogenic factors, and growth factors [9,56,57]. This is another argument for the involvement of neutrophils in different clinical conditions, like acute inflammatory diseases, wound healing, autoimmune diseases, and cancer [9,56,57]. Neutrophils produce cytokines via degranulation, protein synthesis, NET-associated cytokine release, and the expression of receptor-bound cytokines [58].

## 4. Neutrophil-to-Lymphocyte Ratio (NLR)

The NLR (ratio between the neutrophil and lymphocyte counts in peripheral blood) is a newly described biomarker, linking innate (the first-line neutrophils) and adaptive immunity (lymphocytes), through numerous relationships, with different diseases like sepsis, cancer, myocardial infarction, etc. [59,60,61], as well as with basically with any condition associated with tissue damage and inflammation that is ultimately followed by SIRS [59]. Even if a cut-off value is not definitely established and different values have been reported to be of reference (Figure 1), the significance of the NLR is high in many clinical conditions; most of the studies agree that an NLR value above three significantly increases mortality and worsens prognosis [59,61,62].

Of particular interest for anesthesia is the value of the NLR as a biomarker in sepsis and cancer and during the perioperative period. It has been demonstrated that the NLR is an early biomarker for sepsis and prognosis in sepsis, especially in those patients without leukocytosis [59,62,63].

In cancer biology, inflammation plays an important role in tumorigenesis, while in turn, cancer, once initiated, maintains inflammation due to tumor-released pro-inflammatory cytokines, subsequently recruiting immune cells [59,64]. This is why the NLR has been proved to be an early and reliable biomarker for cancer-related inflammation and prognosis, especially in solid tumors like hepatocarcinoma or colorectal cancer [59,65,66]. Moreover, it has been shown that the NLR may predict OS and DFS in surgical cancer patients.

In surgery, the preoperative value of the NLR was proved to be correlated with postoperative complications and mortality for different types of surgical interventions [59,67,68].

### The Role of Neutrophils in Cancer

Despite many years of research, it is only in the last few years that the role of neutrophils in cancer has been extensively studied and begun to be understood.

It was shown in the previous sections that inflammation plays an important role in tumorigenesis [69]. Inflammation, especially chronic inflammation, acts as a promoter for transforming subthreshold neoplastic states into tumors [69]. Once developed, the tumor produces cytokines and chemokines that attract different leukocyte populations: neutrophils, macrophages, lymphocytes, and mast cells. These cells produce pro-inflammatory cytokines (of particular interest are TNF-α, IL-6, and interferon), proteases, MMPs, and NETs that support inflammation and favor tumor development and metastasis. Monocytes attracted to the tumor site turn into tumor-associated macrophages (TAMs) that play the role of a two-sided coin: on one side, they kill tumor cells, and on the other side, they produce a number of angiogenic (VEGF-C, D, F) and lymphangiogenic growth factors, cytokines, and proteases involved in tumor biology and progression [69,70].

The same double role in tumorigenesis is held by neutrophils. There is growing evidence that neutrophils play an important role in cancer biology and tumor progression [59,64]. It has been reported that large numbers of tumor-associated neutrophils (TANs), as well as an increased NLR, are associated with advanced stages of cancer, incomplete surgical resection, and poor short- and long-term postoperative prognosis [59,65,66,71]. The pro-tumor functions of neutrophils include ROS, MMPs, cytokines, and chemokines [71]. MMPs (especially MMP9) stimulate tumor cell proliferation and the production of VEGF and subsequent angiogenesis, while IL-6, IL-1β, and TNF-α, among pro-inflammatory cytokines, support chronic inflammation preceding or associated with tumor [72]. ROS are involved in the genotoxicity in tumor cells promoting carcinogenesis and neutrophil elastases promoting tumor cell proliferation [72,73]. Neutrophils also stimulate metastasis via a number of mechanisms including NETosis, promoting the retention of circulating tumor cells via ICAM clustering or direct adhesion [72]. Another mechanism implicated in the pro-tumor effects of neutrophils is represented by gMDSCs (also called PMN-MDSCs). These are low-density neutrophils with a suppressive and pro-tumoral phenotype that become detectable in advanced stages of cancer and are associated with T-cell impairment and immune suppression [74]. These cells play a role in tumor progression by promoting tumor vascularization, neoplastic cell local invasion, and distant metastasis and via the regulation of T and NK cells’ anti-tumor activity. In practice, MDSCs form a pre-metastatic niche, allowing the invasion of disseminated cancer cells [74,75]. These cells are targets for therapeutic modulation in cancer patients.

The other side of the coin represents the anti-tumor effects of neutrophils. There is clear evidence that neutrophils have a number of anti-tumor effects. Thus, neutrophils may directly kill cancer cells when activated by different mechanisms including ROS (inducing tumor cell lysis in early stages or rapidly growing tumors), MMPs (MMP8), direct lysis and apoptosis, and the regulation of T-cell functions [67,76].

The role of inflammation in cancer development and progression is also clinically supported by the anti-tumor effects of NSAIDs and aspirin that will be discussed later in this review.

## 5. The Influence of Anesthetic Drugs on Neutrophils and Their Functions

Over the years, a number of studies have reported that anesthetic agents may influence neutrophils and their functions (Figure 2).

### 5.1. Intravenous Anesthetic Agents and Neutrophils

#### 5.1.1. Propofol

Propofol (2,6-diisopropylphenol) is, probably, the most widely used i.v. agent for both the induction and maintenance of anesthesia (total intravenous anesthesia—TIVA). TIVA has been extensively used in clinical practice in the last few years for different reasons: environmental protection, quality of recovery, risk of malignant hyperthermia, potential decreased risk of delirium, and surgical reasons (interference of anesthesia with the surgical field). In the last few years, after Wigmore’s study was published [77], a number of publications also reported that TIVA is followed by increased overall survival and disease-free survival after surgery in certain types of cancers, especially digestive cancers [78,79,80].

Propofol acts on GABA A receptors, allowing Cl^−^ ions to enter the cells [81,82]. GABA is one of the most important neurotransmitters in the central nervous systems [83], and GABA A (and B) receptors are widely distributed in neutrophils, monocytes, and macrophages, probably mediating the effect of propofol in immune cells [84]. In higher concentrations, propofol directly opens chloride channels and blocks glutamate release via the inhibition of certain sodium channels [81,85]. Propofol also inhibits the N-methyl-d-aspartate (NMDA) receptors and modulates calcium influx, while muscarinic and dopaminergic receptors have also been involved in propofol’s mechanisms of action [81,86].

In clinically relevant concentrations, older studies reported that propofol inhibited phagocytosis and the killing of E.coli and Staphylococcus aureus [87], while others did not find this effect [88]. More recent publications highlighted propofol’s effects such as regulating macrophage and neutrophil phagocytosis, pyroptosis, the production of pro-inflammatory cytokines (especially IL-6), and the inhibition of macrophage migration and recruitment [83,89]. Taking into consideration the effect of propofol on bacterial killing, we may expect an increase in the incidence of infections in ICUs where propofol is used for sedating critical patients for long periods of time. Even if clinical studies are limited, however, the studies published so far have not reported an increase in the incidence of pneumonia or surgical site infections in patients with TIVA compared with inhalation anesthesia, but on the contrary, the incidence was lower in abdominal surgery cases [90,91]. The same lack of significant differences was reported between sedation with propofol and dexmedetomidine in ICUs [92]. What is interesting is the fact that these effects are not mediated in humans by GABA A receptors, which are not elicited on neutrophils’ surfaces but via other mechanisms, like the inhibition of ROS production, p44/42 mitogen-activated protein kinase phosphorylation, and mitochondrial membrane potential [81,83,93,94]. Since most of these studies have been performed in animals, there is a lack of human studies where confounding factors may interfere, especially studies in critically ill patients, so prospective RCTs in critically ill patients for the potential effects of propofol sedation are mandatory.

#### 5.1.2. Ketamine

Ketamine is an i.v. anesthetic agent increasingly used in the last few years mainly due to it having analgesic and antidepressant properties without depressing respiration. Ketamine may also be used as an induction agent (alone or in combination with other agents or opioids) in unstable patients, taking into consideration its hemodynamic effects, especially for increasing blood pressure. Ketamine mainly acts as an antagonist of NMDA receptors, but other targets for its action have been described: the GABA A receptor, cyclic nucleotide-gated subtype 1 receptor (hyperpolarization-activated), and α- and β-adrenergic, -muscarinic, and -opioid receptors [81,95,96,97].The few studies of ketamine’s effects on neutrophils showed that ketamine attenuated the superoxide anion production of human neutrophils via PMA (phorbol myristate acetate) stimulation and decreased oxidative burst (via decreased phosphorylation of p47) and neutrophil adhesion in a concentration-dependent manner [81,98]. However, further studies are necessary in the context of the increased use of ketamine as an adjunct to anesthesia and as an antidepressant.

#### 5.1.3. Benzodiazepines

Benzodiazepines act by potentiating the inhibitory effect of GABA at the postsynaptic site, resulting in the inhibition of neuronal membrane via hyperpolarization. Some benzodiazepines act on GABA A peripheral benzodiazepine receptors. Studies of the effect of benzodiazepines in human neutrophils are limited and show contradictory effects. Most of the studies showed the inhibition of fMLP-induced human neutrophil chemotaxis and superoxide production (for diazepam) [81], reduced oxidative burst, and reduced CD11b/CD18 expression and p38 mitogen-activated protein kinase phosphorylation in neutrophils (for midazolam) [81,99]. However, Marino et al. reported increased human neutrophil migration and phagocytosis via an interaction with the peripheral benzodiazepines’ receptors for diazepam [100]. Diazepam appeared to enhance phagocytosis by increasing intracellular calcium, while the stimulatory effect on migration was not calcium-dependent [100].

#### 5.1.4. Alpha-2 Agonists

α2-adrenergic receptor agonists stimulate α2 adrenergic receptors, and in the case of clonidine, central imidazoline receptors are added as targets. More recently, dexmedetomidine has been commonly used for the sedation of critically ill patients in ICUs (in which case, it is administered for up to 72 days, or for even more days in some cases) or as an adjunct to anesthesia. Both dexmedetomidine and clonidine have analgesic effects and decrease opioid use. The treatment of delirium or alcohol withdrawal are other indications. Neutrophils express adrenoreceptors (including α2) on their surfaces, so an interaction with α2 agonists is expected at least due to this target [101]. A number of mostly in vitro studies in the last few years have shown that dexmedetomidine suppressed pro-inflammatory cytokines (IL-6, TNF-α, Necrosis factor) and antimicrobial effectors (ROS, RNS, NO, iNOS), as well as the local aggregation of neutrophils, leading to an in vitro anti-inflammatory effect that may be beneficial in critically ill patients in states with increased inflammatory response (SIRS) [102]. A number of pre- and mid-clinical studies suggest that dexmedetomidine may have anti-tumor effects by having indirect anti-inflammatory effects and directly inhibiting the survival, proliferation, and invasion of tumor cells [103].

Further clinical studies are necessary to quantify the impacts of these effects on long-term postoperative outcomes in cancer patients.

#### 5.1.5. Inhalation Agents and Neutrophils

Inhalation agents act via numerous mechanisms, including increasing the sensitivity of GABA A receptors [81,104], potentiating the effects of glycine (an inhibitory transmitter) on glycine receptors, inhibiting α4β2 nicotinic acetylcholine receptors, and interacting with ion channels including the family of neurotransmitter receptors [105,106]. Volatile anesthetics impaired neutrophil functions by decreasing the number of reacting PMNs, chemotaxis and adhesion, and ROS production [107,108,109]. However, there are studies reporting contrary results, i.e., increased phagocytosis of apoptotic neutrophils by macrophages, thus reducing lung inflammatory injury after isoflurane exposure or having no impact on phagocytosis after isoflurane exposure [110,111]. A number of mostly animal model studies, subsequently focused on the effects of volatile agents on inflammation after ischemia/reperfusion. It has been reported that inhalation agents decreased neutrophil infiltration and subsequent organ injury and TNF-α and IL-6 levels [107,108,109,112]. It was also shown that inhalation agents suppress NK cells’ activity, with potential implications for immune surveillance and immune suppression [113,114,115] after cancer surgery, but data on these effects’ impacts on clinical long-term outcomes are still missing.

#### 5.1.6. Local Anesthetics (LAs)

LAs are used mostly in the context of regional anesthesia. However, in the last few years, the i.v. infusion of lidocaine has gained popularity due to its multiple effects. It has been demonstrated that regional anesthesia/analgesia reduces stress response to surgery and the need for opioids while increasing NK cells’ activity and the number of T-helper cells.

LAs act by blocking voltage-gated sodium channels (VGSCs), thus blocking the transmission of the neural signal [116]. VGSCs are a family composed of nine members (nine subtypes defined by the nine types of α subunit) and are common not only in nervous systems but also in other types of cells, like heart muscles, pancreatic cells, and cancer cells. This is why VGSCs and LAs have many clinical implications in cardiac and respiratory disorders, diabetes, cognitive dysfunctions, and cancer [117].

Recent studies have shown that lidocaine, in vitro and in humans (as well as in Kulińska’s study of ropivacaine), enhanced NETosis in neutrophils at clinically relevant concentrations, thus having the possibility to change the course of inflammation and metastasis [118,119]. Kulinska et al. also showed that LAs upregulate nitric oxide synthase (NOS) production by neutrophils, thus increasing NO synthesis [120]. Moreover, a recent meta-analysis showed that perioperative i.v. lidocaine can be recommended as an anti-inflammatory and analgesic for elective surgery [121]; the same effects were reported in emergency departments for different painful inflammatory conditions [122]. It was shown that after elective surgery, lidocaine decreased IL-6, TNFα, and IL-1RA levels [121]. Further studies are also needed to better define the roles of LAs and, in particular, i.v. lidocaine as anti-inflammatory and analgesic drugs.

#### 5.1.7. Opioids

Opioids are used perioperatively and in painful conditions for analgesia. These drugs exert their effect by acting on G-coupled protein µ-, δ-, and κ-receptors. Opioid receptors are widely expressed in the central and peripheral nervous systems and non-neuronal cells, like the gastrointestinal tract or immune cells [123]. µ-opioid receptors and nociceptin–orphanin receptors have been described in polymorphonuclear cells and NK, T, and B cells [123]. These receptors can be activated not only by opioids but also by cytokines (IL-1, IL-2, IL-7, IFN, TNF) and other inflammatory mediators. This is the pharmacological rationale for which, besides the well-known side effects of opioids mediated via central opioid receptors (e.g., respiratory depression), immune suppression is another effect, mediated by peripheral opioid receptors. In the last few years, it has been shown that morphine and other opioids are associated with an increased risk of infection and sepsis [124]. Morphine and other opioids inhibit neutrophil functions including phagocytosis, migration, antibody, and interleukin/chemokine production, thus influencing immune response during infections or cancer [125]. In cancer, on the other hand, certain neutrophils release endogenous opioids that diminish pain and inflammation, thus having potential negative effects on tumor progression [126]. Moreover, there are differences between opioids in immune effects and effects on cancer progression, with morphine probably being the most immune-suppressing opioid, while fentanyl has some immune stimulatory effects on regulatory T cells [127]. Further studies are very much needed to identify if these differences between opioids have clinical significance in cancer patients.

### 5.2. Minor Analgesics and Neutrophils

#### 5.2.1. NSAIDs/Aspirin

NSAIDs are used during the perioperative period for analgesia. Their main mechanism of action is the inhibition of cyclooxygenase 1 and 2 (COX), enzymes involved in prostaglandin synthesis; thus, NSAIDs inhibit prostaglandin synthesis. Other mechanisms of action include interference with cell adhesion, involving cancer cells, platelets and leukocytes adhesion; the generation of superoxide anion in the plasma membranes of leukocytes, and the inhibition of neutrophil elastase [128]. On the other hand, NSAIDs have antineoplastic and neuroprotective effects [129]. Several types of cancers overexpress COX with subsequent implications for tumorigenesis [127,130]. Some of the NSAIDs inhibit transcription factors like NF-kappa B, influence neutrophil chemotaxis, or regulate signaling pathways [128], more recently described as other mechanisms for the anti-inflammatory effects of NSAIDs [127,128].

#### 5.2.2. Nefopam

This newly introduced centrally acting non-opioid analgesic mainly acts by inhibiting the synaptic uptake of several transmitters like dopamine, noradrenaline/norepinephrine, and serotonin. Recently, it has been shown that nefopam inhibits neutrophil migration and wound-induced neutrophil recruitment, thus having anti-inflammatory effects, even if it is not an anti-inflammatory drug initially [131].

#### 5.2.3. Impacts on Short- and Long-Term Outcomes

Taking into consideration the effects of anesthetic agents not only on neutrophils but also on immune response and cancer cell biology, the question is whether these effects impact the short- and long-term outcomes of surgical patients. At the moment, there are not many definitive conclusions on the impact of anesthetic agents on patients’ outcomes, and most of the studies are only under way. However, regarding regional anesthesia and postoperative long-term outcomes in cancer patients, most of the studies did not find a significant impact of regional anesthesia on cancer mortality and the incidence of recurrence [132,133].

The other question is whether TIVA may ameliorate long-term outcomes in cancer patients compared with inhalation anesthesia. While most of the studies in breast cancer patients did not find any difference [133], for other types of cancer (e.g., digestive cancers), there may be a difference in outcome, and some of the RCTs under way (NCT02786329, NCT04188158, NCT00684229, NCT01975064, NCT06017141) may bring light on this issue [133,134]

Preclinical and clinical studies published in the last few years on i.v. lidocaine are encouraging regarding short-term outcomes (pain, inflammation) in cancer patients, taking into consideration its anti-inflammatory and anti-tumor effects, so another question is whether i.v. lidocaine may impact disease-free survival and OS in cancer patients undergoing surgery.

At the moment, there are no data on this topic, but the above-mentioned RCTs under way may also generate more data on this issue.

Last but not least, another intriguing the question is whether anesthetics may, on the other hand, promote the pro-tumorigenic activity of neutrophils. To our knowledge, at this moment, there are no data on this aspect, so further studies will have to focus not only on the anti-tumor effects of certain anesthetic agents/techniques but also on the effects of promoting the pro-tumorigenic effects of certain neutrophil subpopulations.

### 5.3. Anesthetic Drugs and the NLR

As shown before, the NLR is a newly described biomarker closely related to patient outcomes. The NLR has many implications for anesthesia in sepsis and cancer and predicting postoperative complications [59,67,68]. As a consequence, a number of studies aimed to investigate whether anesthetic drugs may influence the NLR. In a recent study, Hasselager reported that the NLR was significantly higher during propofol anesthesia and on day 1 compared with sevoflurane in healthy volunteers who did not undergo surgical intervention [79]. However, Ní Eochagáin reported that a propofol–paravertebral block slightly attenuated the increase in the NLR in breast cancer surgery, with no benefit for long-term outcomes [135].

I.v. lidocaine postoperatively decreased the NLR after breast cancer surgery [136], while Alexa et al. did not find an influence of i.v. lidocaine at 24 h after colorectal surgery [137]. An erector spinae plane block (ESPB) significantly decreased the NLR after spine surgery [138]. The impact of the influence of anesthetic agents on the NLR for patient outcomes still needs to be determined via future RCTs studying long-term outcomes. Such studies should follow patients for a long time postoperatively to assess NLR variations’ relationship with clinical outcomes.

### 5.4. Anesthetic Agents and the Efficacy of Anti-Cancer Drugs Administered after Surgery

There is a paucity of data on the potential interactions between anesthetic agents and anti-cancer drugs, namely chemotherapeutical agents and immune therapy. The interaction between anesthetic agents and anti-cancer drugs may be produced via an indirect mechanism of modulation of the host immune response and direct interference with cancer cell biology. The few mostly in vitro and animal model studies published so far showed that propofol may increase chemotherapeutic activity via the inhibition of Slug expression (increasing paclitaxel-induced apoptosis), abrogation of gemcitabine-induced DNA-binding activity of NF-kB, and downregulation of NF-kB [139,140]. However, there are studies showing that propofol may decrease chemotherapeutic efficiency via the inhibition of gap junction function [139,140,141].

Lidocaine may attenuate chemotherapeutic drug resistance or increase chemotherapeutic drug efficiency by decreasing the expression of miR-21 or via the downregulation of ABC transporter protein expression [139]. Animal model studies also showed that lidocaine decreased tumor proliferation and increased chemotherapeutic drug efficiency via the activation of caspase 3, decreased Bcl-2 and Bax expression, and the inactivation of ERK1/2 and p38 pathways [142]. Future studies are needed to determine the clinical significance of these results in terms of ameliorating long-term outcomes.

## 6. Conclusions

Neutrophils are key factors in defense against infection and cancer and, in consequence, are important in onco-anesthesia and the perioperative period. Although many factors intervene during this period, there are strong arguments that anesthetic agents may influence neutrophils and their functions. Whether this influence has a meaningful impact on the short- and long-term outcomes of surgical patients remains to be determined in future large RCTS, which are urgently needed. At the moment, there are insufficient data to justify choosing certain anesthetic techniques/drugs to ameliorate postoperative outcomes in cancer patients.

## Figures and Tables

**Figure 1 ijms-25-04033-f001:**
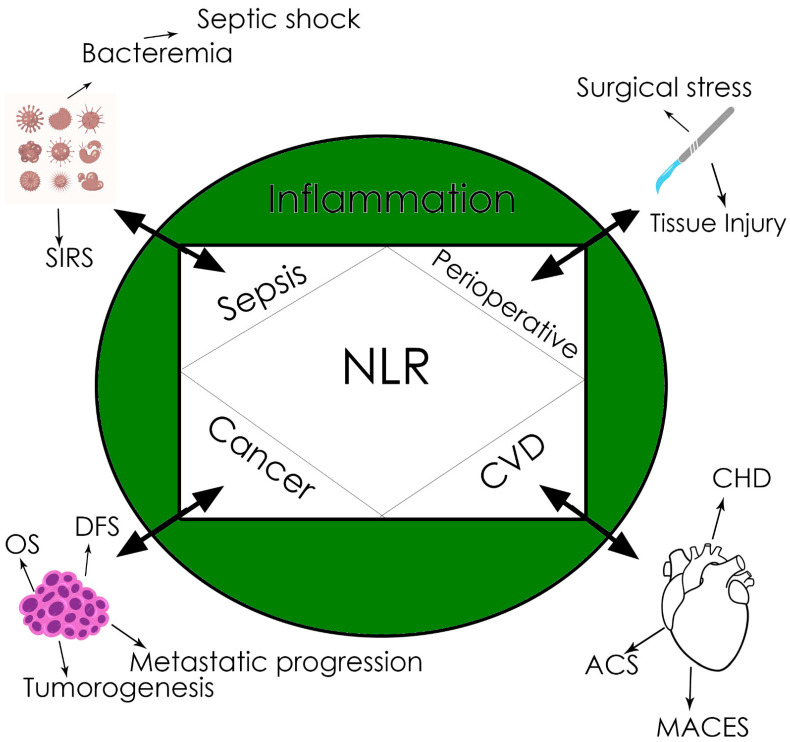
NLR’s clinical implications. CVD, cardiovascular disease. CHD, coronary heart disease. ACS, acute coronary syndrome. MACEs, major adverse cardiac events. DFS, disease-free survival. OS, overall survival. SIRS, systemic inflammatory response syndrome.

**Figure 2 ijms-25-04033-f002:**
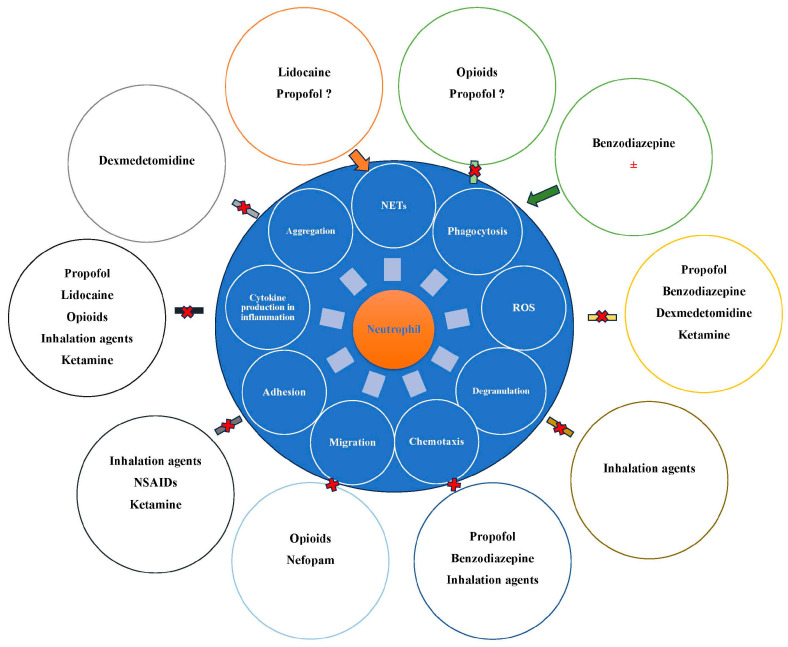
Influence of anesthetic drugs on neutrophils. ROS, reactive oxygen species. NETs, neutrophil extracellular traps. AINS, nonsteroidal anti-inflammatory drugs. Cross marks suggests inhibition, while arrows suggests stimulation.

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
