# Peer review of "Neutrophils and Anesthetic Drugs: Implications in Onco-Anesthesia"

_ijms, 2024, doi:10.3390/ijms25074033_

Round 1

Reviewer 1 Report

Comments and Suggestions for Authors

Content suggestions:

1.         Can the Authors add some details about the visualization of NETs and its use in monitoring of the presence and quantity of such NETs in the patients ?

2.         I would like to kindly ask the Authors to add some information about the possibilities of the modification of the treatment in oncological patients who need to undergo anesthesia.

From my point of view, after the implementation of the responses to the questions of the reviewers in the form of minor revision, the manuscript might be published.

Author Response

Reviewer 1

Suggestions:

  1. Can the Authors add some details about the visualization of NETs and its use in monitoring of the presence and quantity of such NETs in the patients ?

Response: Thank you for suggestion. We added a section with the following paragraphs on NET asessment

“2.4.2. Assessment NET Formation

Several methods are described for visualization and quantification of NETs. They are classified in direct and indirect methods.

Immunofluorescence microscopy is a direct qualitative method consisting in granular staining for neutrophil-associated enzymes, histones and extracellular DNA, by using specific dyes and immunostaining for the enzymatic components of NETs [35,36]. This method allows NETs identification in several biological cell culture mediums, tissue sections, peripheral blood and bronchoalveolar lavage. Having the fact that it can differentiate different stages of NETs formation and it allows differentiation between NETs apoptosis and necrosis [35]. However, it is operator-dependent, time-consuming, laborious and does not allow a quantitative analysis [36]. Nowadays a semi-automatic quantitative assessment of NETosis is possible, using specific antibodies against histone, chromatin, cleavage site of histone H3 [35,37].

Electron microscopy and scanning electron microscopy (SEM) consist in electron beam passing through or reflecting from the surface of the sample[35]. Which is then projected on fluorescence surface or is collected by a sensor that create an image [38]. SEM is first choice for visualizing NET formation. It creates a better image compared with simple electron microscopy [37]. The disadvantage is determined by the possibility of the electron beam to stimulate the formation of NETs [37].

In-vivo and in-situ methods use intravital two-photon microscopy and subsequent electron emission detection based on DNA components [35,36]. Recent data show that NETs are less expressed in vivo compared to in vitro [37].

Real time NETosis assessment is done by IncuCyte ZOOM imaging platform. This platform uses cell membranes properties with permeability of DNA dyes (NUCLEAR-ID and Sytox Green). By detecting morphological changes in cells and nuclei this method can identify NETs from other cell death but has the disadvantage of false positive results if cell membrane integrity is compromised [39].

Flow cytometry and multispectral image flow cytometry allows measurement of scattered light or fluorescence signals emitted by properly irradiated cells leading to the differentiation of distinct cell populations in whole blood. It uses specific dyes (Sytox Green) or conjugated antibodies, which bind to cytokines, receptors, cellular components and DNA [35]. Multispectral image flow cytometry can measure in whole blood both “suicidal” and “vital” NETs and it can quantify NETs in the early stages that underestimate the formation of NETosis at a longer time interval [35,37].

Enzyme-Linked Immunosorbent Assay (ELISA) quantifies some products of neutrophil degradation characteristic for NETs without direct visualization [35,37]. Although, there is a controversy on the best marker for detecting NET formation [35]. CitH3-DNA ELISA is validated for reliable NET quantification in human plasma. While MPO-DNA complexes may also be identified by ELISA, discrepancies between in vivo and in vitro values have been identified [35].

Western blot is another indirect method, using specific antibodies to quantify a protein involved in the formation of NETs. It is a very sensitive and specific method with some disadvantages related costs, the loose of specificity in a mix of protein antibodies [35]. “

  1. I would like to kindly ask the Authors to add some information about the possibilities of the modification of the treatment in oncological patients who need to undergo anesthesia.

            Response: Thank you for your suggestion. We added in conclusions that at the moment there are no sufficient data to justify choosing a specific anesthetic technique to improve outcome in cancer patients. A paragraph on the Impact on short and long-term outcome was also added as a response to Reviewer 2 suggestions.

Reviewer 2 Report

Comments and Suggestions for Authors

This review describes the role of anesthetic drugs on the activity of neutrophils and it discuss how these anesthetics can influence the prognosis of cancer patients undergoing surgery.

There are several aspects that need to be clarified:

1) In the paragraph 4.1 authors claims the double role of neutrophils in tumorigenesis, reporting that neutrophils can have a pro-tumorigenic activity. One immune-subset that play a role in promoting tumor progression are gMDSCs. These immune-suppressor cells derive from neutrophils and in the TME play a role in inhibiting anti-tumor immune response and promoting cancer progression and metastasis. In this paragraph, authors need to include information about the role of gMDSCs in the TME, because most of the "pro-tumorigenic" activity of neutrophils is due to their differentiation in gMDSCs. 

2) In this review authors describe different effects of different anesthetic on neutrophils, but what is the implication of these anesthetics in the prognosis of cancer patients? It is not clear if the effect of these anesthetics has a long-term effect or short-term effect on immune system. Did these anesthetics impair anti-tumor activity of neutrophils and other immune cells in long-term after their use? Did these anesthetics promote pro-tumorigenic activity of neutrophils? If yes in which types of cancers? NLR is a parameter that needs to be followed up pre and post anesthesia for long time to see if there is a correlation between anesthetics and fluctuation of NLR in cancer patients? The role of anesthetics in affecting neutrophils activity can affect also the efficacy of anti-cancer drugs administered after surgery?

Authors need to answer to these questions to render the manuscript suitable for publication

Author Response

Reviewer 2

This review describes the role of anesthetic drugs on the activity of neutrophils and it discuss how these anesthetics can influence the prognosis of cancer patients undergoing surgery.

There are several aspects that need to be clarified:

  • In the paragraph 4.1 authors claims the double role of neutrophils in tumorigenesis, reporting that neutrophils can have a pro-tumorigenic activity. One immune-subset that play a role in promoting tumor progression are gMDSCs. These immune-suppressor cells derive from neutrophils and in the TME play a role in inhibiting anti-tumor immune response and promoting cancer progression and metastasis. In this paragraph, authors need to include information about the role of gMDSCs in the TME, because most of the "pro-tumorigenic" activity of neutrophils is due to their differentiation in gMDSCs. 

Response: Thank you for our observation. We added a paragraph on gMDSCs.

“Another mechanism implicated in pro-tumor effects of neutrophils is represented by gMDSC (also called PMN-MDSCs). These are low-density neutrophils with a suppressive and pro-tumoral phenotype that become detectable in advanced stages of cancer and are associated with T cell impairment and immune suppression [76]. These cells play a role in tumor progression by promoting tumor vascularization, neoplastic cell local invasion and distant metastasis and by regulation T and NK cells anti-tumor activity. Practically, MDSC form a pre-metastatic niche allowing invasion of disseminated cancer cells [76,77]. These cells are targets for therapeutic modulation in cancer patients.”

  • In this review authors describe different effects of different anesthetic on neutrophils, but what is the implication of these anesthetics in the prognosis of cancer patients? It is not clear if the effect of these anesthetics has a long-term effect or short-term effect on immune system. Did these anesthetics impair anti-tumor activity of neutrophils and other immune cells in long-term after their use? Did these anesthetics promote pro-tumorigenic activity of neutrophils? If yes in which types of cancers? NLR is a parameter that needs to be followed up pre and post anesthesia for long time to see if there is a correlation between anesthetics and fluctuation of NLR in cancer patients?

Response: Thank you for this very good suggestion. We added the following paragraph on the influence of anesthetic agents and techniques on short and long term outcome in cancer patients.

5.2.3. Impact on short- and long-term outcome

Taking in consideration the effects of anesthetic agents not only on neutrophils but on immune response and cancer cells biology, the question is if these effects impact short and long term outcome of surgical patients. At the moment there are not many definitive conclusions on the impact of anesthetic agents on patients' outcome and most of the studies are on the way. However, regarding regional anesthesia and postoperative long- term outcome in cancer patients, most of the studies did not find a significant impact of regional anesthesia on cancer mortality and incidence of recurrences [134,135].

The other question is if TIVA may ameliorate long term outcome in cancer patients when compared with inhalation anesthesia. While most of the studies in breast cancer patients did not find any difference [135] for other types of cancer (e.g. digestive cancers) there may be a difference in outcome and some of the RCTs on the way (NCT02786329, NCT04188158, NCT00684229, NCT01975064, NCT06017141) may bring light on this issue [135,136] 

Preclinical and clinical studies published in the last years on i.v. lidocaine are encouraging regarding short-term outcome (pain, inflammation) in cancer patients taking in consideration its anti-inflammatory and anti-tumor effects, so another question is if i.v. lidocaine may impact disease free survival and OS in cancer patients undergoing surgery.

At the moment there are no data on this but the above mentioned RCTs on the way may also bring more data into this issue.

Last but not least, another intriguing the question is if anesthetics may, on the other side, promote pro-tumorigenic activity of neutrophils. To our knowledge, at this moment there are no data on this aspect so further studies will also have to focus not only on anti-tumor effects of certain anesthetic agents/techniques, but contrary on those effects of promoting the pro-tumorigenic effects of certain neutrophils subpopulations.

Short phrases on the influence of specific anesthetic agents on the outcome have been also added as follows:

5.1.4. Alpha-2 agonists: “A number of pre- and clinical studies suggest that dexmedetomidine may have anti-tumor effects by indirect anti-inflammatory effects and directly inhibiting survival, proliferation and invasion of tumor cells [105].”

5.1.5. Inhalation agents and neutrophils: “It was also shown that inhalation agents suppress NK cells activity with potential implications on immune surveillance and immune suppression [115,116,117] after cancer surgery but data on these effects on clinical long term outcome are still missing.”

5.3. Anesthetic drugs and NLR

“The impact of the influence of anesthetic agents on NLR on patient’s outcome still need to be determined by future RCTs on long term outcome. Such studies should follow patients for a long time postoperatively to both assess NLR variations in relationship with clinical outcome.”

“5.4. Anesthetic agents and the efficacy of anti-cancer drugs administered after surgery

There is a paucity of data on the potential interactions between anesthetic agents and anti-cancer drugs, namely chemotherapeutical agents or immune therapy. The interaction between anesthetic agents and anti-cancer drugs may be produced by an indirect mechanism of modulation of host immune response and a direct interfering with cancer cell biology. The few mostly invitro and animal model studies published so far showed that propofol may increase chemotherapeutic activity by inhibition of Slug expression (increasing paclitaxel-induced apoptosis), abrogation of gemcitabine-induced DNA-binding activity of NF-κB and downregulation of NF-kB [141,142]. However, there are studies showing that propofol may decrease chemotherapeutic efficiency by inhibition of gap junction function [141,142,143].

Lidocaine may attenuate chemotherapeutic drug resistance or may increase chemotherapeutic drugs efficiency by decreasing expression of miR-21 or by downregulation of ABC transporter protein expression [141]. Animal model studies also showed that lidocaine decreased tumor proliferation and increased chemotherapeutic drug efficiency by activation of caspase 3, decreased Bcl-2 and Bax expression, inactivation of ERK1/2 and p38 pathways [144]. Future studies are needed to determine the clinical significance of these results in terms of ameliorating long-term outcome.”

  1. Conclusions

“At the moment there are no sufficient data to justify choosing certain anesthetic technique/drugs to ameliorate postoperative outcome in cancer patients.”

  • The role of anesthetics in affecting neutrophils activity can affect also the efficacy of anti-cancer drugs administered after surgery?

Response. Thank you. A paragraph on this has been added.

5.4. Anesthetic agents and the efficacy of anti-cancer drugs administered after surgery

There is a paucity of data on the potential interactions between anesthetic agents and anti-cancer drugs, namely chemotherapeutical agents or immune therapy. The interaction between anesthetic agents and anti-cancer drugs may be produced by an indirect mechanism of modulation of host immune response and a direct interfering with cancer cell biology. The few mostly in vitro and animal model studies published so far showed that propofol may increase chemotherapeutic activity by inhibition of Slug expression (increasing paclitaxel-induced apoptosis), abrogation of gemcitabine-induced DNA-binding activity of NF-κB and downregulation of NF-kB [141,142]. However, there are studies showing that propofol may decrease chemotherapeutic efficiency by inhibition of gap junction function [141,142,143].

Lidocaine may attenuate chemotherapeutic drug resistance or may increase chemotherapeutic drugs efficiency by decreasing expression of miR-21 or by downregulation of ABC transporter protein expression [141]. Animal model studies also showed that lidocaine decreased tumor proliferation and increased chemotherapeutic drug efficiency by activation of caspase 3, decreased Bcl-2 and Bax expression, inactivation of ERK1/2 and p38 pathways [144]. Future studies are needed to determine the clinical significance of these results in terms of ameliorating long-term outcome.

Round 2

Reviewer 2 Report

Comments and Suggestions for Authors

Authors addressed my comments in a proper manner, including important additional information in this review.

In my opinion, the review is suitable for publication now